# Cultivable Bacteria Associated with the Microbiota of Troglophile Bats

**DOI:** 10.3390/ani12192684

**Published:** 2022-10-06

**Authors:** Maria Foti, Maria Teresa Spena, Vittorio Fisichella, Antonietta Mascetti, Marco Colnaghi, Maria Grasso, Chiara Piraino, Franco Sciurba, Rosario Grasso

**Affiliations:** 1Department of Veterinary Science, University of Messina, Via Palatucci 13, 98168 Messina, Italy; 2Department of Biological, Geological and Environmental Sciences, University of Catania, Via Androne 81, 95124 Catania, Italy; 3Centre for Mathematics and Physics in the Life Sciences and Experimental Biology (CoMPLEX), Department of Genetics, Evolution and Environment, University College London, 610 Darwin Building, Gower Street, London WC1E 6BT, UK; 4Zooprophylactic Institute of Sicily, Via Gino Marinuzzi 3, 90129 Palermo, Italy

**Keywords:** bats, bacteriological test, Gram-negative bacteria, Gram-positive bacteria, public health, Southern Italy

## Abstract

**Simple Summary:**

Troglophile bats live in colonies, often in sites exploited for agro-pastoral purposes. Determining the composition of the microbiome of bats is an important step in understanding their ecology and biology and can also provide information on the spread of pathogenic bacteria in their populations. This study aimed to determine how epidemiological factors shape the microbiome of troglophile bats and evaluate the occurrence of potentially pathogenic bacterial species. A total of 413 Gram-negative and 183 Gram-positive strains were isolated from 189 individuals of four species of troglophile bats living in Sicilian and Calabrian territory (Italy). Besides few potentially pathogenic bacteria, several strains with a hypothesized symbiotic role were found.

**Abstract:**

Background: The study of bats is of significant interest from a systematic, zoogeographic, ecological, and physiological point of view. The aim of this study is to investigate the culturable aerobic enteric, conjunctival, and oral bacterial flora of bats to determine their physiological microbiome and to investigate the possible occurrence of pathogenic bacteria. Methods: Five hundred and sixty-seven samples were collected from 189 individuals of four species of troglophile bats (*Myotis myotis*, *Myotis capaccinii*, *Miniopterus schreibersii*, and *Rhinolophus hipposideros*) living in Sicilian and Calabrian territory (Italy). All samples were tested for Gram-negative bacteria; conjunctival and oral swabs were also submitted to bacteriological examination for Gram-positive bacteria. Results: Four hundred thirteen Gram-negative strains were isolated. Of these, 377 belonged to 17 different genera of the family Enterobacteriaceae and 30 to five other families. One hundred eighty-three Gram-positive strains were isolated. Of these, 73 belonged to the Staphylococcaceae family, 72 to the Bacillaceae family and 36 to four other families. Besides some potentially pathogenic strains, several bacterial species have been found that are common to all the bat species studied. These could perhaps play a physiological or nutritional role. Conclusion: A great variety of bacterial species were identified in the cultivable microbiota of southern-Italian troglophile bats, including several potentially pathogenic strains and numerous putatively symbiotic species.

## 1. Introduction

In Europe, the Chiroptera represent the mammalian order with the largest number of species, but almost all of them are threatened or at extinction risk. For this reason, the list of strictly protected species (Annex II) by the “Berne Convention on the Conservation of European Wildlife and Natural Habitats” (1979) includes all Microchiroptera, with the exception of *Pipistrellus pipistrellus*. The species referable to the Italian territory are currently 35 [1]. They belong to the Italian fauna virtually all European species divided into three families: Rhinolophidae, Vespertilionidae, and Molossidae. To these is added the family Miniopteridae, recently considered as distinct from that of the Vespertilionids [2]. All Italian Chiroptera feed on Arthropods and are nocturnal even if several species come out of shelters in the presence of light [1]. The diet of *Myotis capaccinii* also includes aquatic larvae of Diptera and fish fry [1].

Interest in bats is growing exponentially as a result of their alleged involvement in human and domestic animals’ viral diseases emergence, including severe acute respiratory syndrome (SARS), Middle East respiratory syndrome (MERS), disease of Nipah and Hendra [3,4], rabies [5], and, more recently, CoV disease 2019 (COVID-19) [6]. Furthermore, an increasing number of other viruses, whose pathogenic potential is still unknown, is constantly being reported in bats globally [7,8]. Compared to a great number of studies conducted on viral agents, the possible role of bats as potential carriers of pathogenic bacteria has not been explored in great depth, especially in European bats [9,10]. Most studies on the pathogenetic diversity of bats have focused on single pathogens and very little research has been carried out on the European continent. In European bats, infections by *Leptospira* spp. [11], *Borrelia* spp. [12], *Micoplasma* spp. [13], *Bartonella* spp. [10,11,14,15], and *Rickettsia* spp. [16] have been sporadically reported. Analysis of the intestinal microbial flora [17,18], typically carried out by isolation from guano [19], has revealed the presence of single pathogenic species such as *Campylobacter* spp. [20] and *Salmonella* spp. [9]. Data on the occurrence of Gram-positive bacteria are even scarcer, and most studies have focused on enteric bacteria from genera *Enterococcus*, *Lactococcus*, and *Lactobacillus* [17,21]. Walther et al. (2008) [22] describe the isolation of a methicillin-resistant *Staphylococcus aureus* strain from a bat wound. Vandžurová et al. (2013) [19] isolated high percentages of *Staphylococcus nepalensis* in the guano of *Myotis* spp. in Slovakia. The aim of this study is to study culturable aerobic bacteria of bats living in caves in Southern Italy to investigate the composition of their microbiota and evaluate the possible occurrence of pathogenic bacteria. As the microbiota can influence the state of health or disease through the modulation of the immune system and the competition with pathogenic bacteria, it is important to determine its normal constitution in bat populations. Moreover, this knowledge can help to identify harmful pathogens that can potentially lead to declines in bat populations. We have considered pathogenic the bacteria belonging to Risk Group 2 or higher, as per the National Institutes of Health (NIH) guidelines [23].

## 2. Materials and Methods

### 2.1. Sampling

From September to November 2018, 567 samples (189 rectal swabs (R), 189 conjunctival swabs (C) and 189 oral swabs (O)) were collected from 189 individuals of four different species of troglophile bats (Table 1).

#### 2.1.1. Study Population

The greater mouse-eared bat *Myotis myotis* (Borkhausen, 1797) is a European-Mediterranean species with a range including Eastern, Southern, and Central Europe (up to Southern England), many Mediterranean islands and Asia Minor. All Italian regions are considered to be included in the range of the species [1]. Colonies present mainly in areas with a high percentage of woods [24]. *M. myotis* lives in forest environments, open pastures, and meadows. Its diet is mostly based on insects caught on the ground. It forages over deciduous woodland edge, open deciduous woodland, and pasture, preying on large, ground-dwelling arthropods such as beetles, crickets, and spiders, gleaning them from the ground. This bat typically roosts in underground sites during the whole year, although Northern populations are also known to dwell in buildings (loft-spaces) during the summer. Occasionally, *M. myotis* can also form small colonies in trees. It is an occasional migrant; the longest recorded movement is 436 km [25].

The long-fingered bat *Myotis capaccinii* (Bonaparte, 1837) is found in the Mediterranean basin, in north-western Africa and on almost all the Mediterranean islands. In Italy it is found throughout the country, especially in karst environments. It has a fragmented range, from the Iberian Peninsula to Asia Minor, Israel, Lebanon, Jordan, Turkey, Iran, and Iraq [24]. It feeds on insects, particularly Trichoptera, Neuroptera, Diptera, and occasionally on small fish caught just above water surfaces or just below them. *M. capaccinii* generally roosts in underground habitats (principally caves) and depends strongly on aquatic habitats, as it forages mainly over wetlands and waterways (including artificial waterbodies, such as canals and reservoirs). It seems to prefer clutter-free water surfaces when foraging, probably because this facilitates the echolocation of preys [26]. It feeds on insects, particularly Trichoptera, Neuroptera, Diptera (Chironomidae), and occasionally on small fish (*Gambusia* spp.) caught just above water surfaces or just below them [24]. Movements between summer and winter colonies are mostly within a distance of 50 km (maximum 140 km) [27].

The Schreiber’s bent-winged bat *Miniopterus schreibersii* (Kuhl, 1817) in Europe is present in all Mediterranean regions, including the major islands. It frequents all Mediterranean habitats but with a preference for areas rich in broad-leaved trees. It typically dwells in karst caves and other underground sites [24]. *M. schreibersii* forages mainly in deciduous woodlands and mature orchards (including olive groves), gardens, along hedgerows separating pastures and riverine forests, and in urban areas. In the Mediterranean area it occasionally forages in grasslands but avoids arable land and maquis. It feeds on insects caught in flight. With a highly specialized trophic regime, it preys mainly on Lepidoptera. Non-flying preys are also reported in its diet. Dipterans were the second most consumed prey. Several taxa of Coleoptera, Neuroptera, Orthoptera, and Trichoptera were also recorded. Prey also included many pest arthropod species [28]. Schreiber’s bat is a migrant species that changes its roosts several times during the year; long-distance movements occur occasionally [27].

Lesser horseshoe bat *Rhinolophus hipposideros* (Bechstein, 1800) is now almost absent in much of central Europe but is still widespread in the Mediterranean countries. Summer roosts (breeding colonies) are found in natural and artificial underground sites in the southern part of the range, and in attics and buildings in the northern part of it. Southern populations have nurseries in caves. In winter, *R. hipposideros* hibernates in underground sites (including cellars, small caves, and burrows). It feeds on small insects (Diptera, Lepidoptera, and Neuroptera), captured either in flight or on the ground (e.g., spiders). Members of this species generally hunt individually, very close to the ground, within and along the edges of broadleaf deciduous woodland (which represents their primary foraging habitat), but also in riparian vegetation, Mediterranean and sub-Mediterranean shrubland. Foraging activities take place nearly exclusively within woodland areas, while open areas are avoided [29,30]. Habitat loss and fragmentation may therefore reduce the number of suitable habitats for the lesser horseshoe bat and pose a threat to this species [31].

*Miniopterus schreibersii* is the only one of the four species to be present in all 6 sampling cavities. The studied caves are occupied starting from March–April and abandoned in July–September or until November in the Grotta dei Pipistrelli (SR), depending on the climate [32].

In all sampling sites, these species share the same emergency points from the refuge and similar hours of activity. Once out of the caves, they occupy different trophic niches.

The sampling was carried out in six caves of the Southern Italy territory (Table 2).

Grotta dei Pipistrelli (Sortino, SR, 37°08′30″ N, 15°01′48″ E, cadastral number: Si SR 3526). Located in a protected area (Natural Reserve “Pantalica, Valle dell’Anapo e Torrente Cava Grande”), this cave is not far from the town of Sortino, in areas where semi-wild breeding is practiced. The area is characterized by a natural plateau, deeply engraved by the Anapo River and the Calcinara stream. Pantalica Nature Reserve (over 37 kmq) was awarded in 2005 as UNESCO World Heritage Site for its history, archaeology, speleology, and landscape (UNESCO, 1992–2019). It consists of various natural and semi-natural environments (riparian forest, woodland, shrubland, grassland, and steppe) along with cultivated land [33], which are essential habitats for many invertebrate and vertebrate communities. Grotta dei Pipistrelli opens on a rocky wall overhanging the Calcinara stream, approximately 10 m from the left bank of the river, in a Miocene formation known as “Calcari di Siracusa”. The karst cavity has a sub-horizontal development with a 7.3% west–east average slope and it has been explored for approximately 260 m: between the entrance of the cave, at 234 m above sea level (a.s.l.), and the ending point (253 m a.s.l). The cave hosts very large colonies of Chiroptera and it represents the biggest nursery roost of this region [32,34]. From 2012 to date, Grotta dei Pipistrelli is the only systematically monitored bat cave in Sicily [32,34]. The maximum number of bats recorded was around 10,000 in 2013. Minioptera and large *Myotis* are the predominant genera, with a lower prevalence of Rhinolophids [32]. Below the entrance flows two streams of water (Anapo and Calcinara) from which the bats drink after emerging from the cave.Grotta Palombara (Melilli, SR, 37°06′22″ N, 15°11′39″ E, cadastral number: Si SR 3536). The Integral Nature Reserve “Grotta Palombara” falls within the area of the Climiti Mountains in the eastern sector of the Ibleo plateau, in the territory of the Municipality of Melilli (Syracuse). The area of the Reserve covers an area of 11.25 hectares and was established in 1998 “in order to protect the most important karst cave in eastern Sicily for its underground development and the complexity of the cavity systems, with a varied cave fauna that includes an important Guanobia component”. A fossil karst cavity that develops for approximately 800 m. Palombara cave is located near the biggest Italian petrochemical plant, known as “Augusta-Priolo-Melilli”. The cave hosts a colony of bats belonging to the species *Myotis myotis*, *Miniopterus schreibersii*, *Rhinolophus*
*euryale*, and *Rhinolophus ferrumequinum*. The maximum number of Chiroptera recorded was approximately 1000 specimens [32]. Human activity, especially the presence of numerous illegal landfills, has significantly degraded the environment around the cave [34].Grotta Chiusazza (Floridia, SR, 37°01′29″ N, 15°09′35″ E, cadastral number: Si SR 3533). The total length of this cave is approximately 250 m. The area, located in south-eastern Sicily, falls within the territory of Syracuse in the “Grotta Perciata-Chiusazza” district on the eastern edge of the Hyblean plateau. The altitudes vary between 200 and 50 m above sea level. The morphology of the area varies from hilly to the west, to sub-flat to the east. The area in which the cave opens is characterized by the presence of intensive monocultures, arable land, and small ponds fed by the runoff from farmland, where the bats go to feed and drink. The cave is populated by a large colony of Minioptera and dozens of Rhinolophids.Grotta del Burrò (Randazzo, CT, 37°49′35″ N, 14°56′04″ E, cadastral number: Si CT 1024). Volcanic cave created as a result of the eruptive activity of Etna. It is a large lava flow tunnel, over 200 m long. Grotta del Burrò develops in prehistoric lava formations of uncertain age (15,000 years–3930 ± 60 years), of which it is not possible to identify the eruptive system. This was probably located a few kilometers south of Monte Spagnolo, today covered by the lava of 1614- 24 (Monte Pomiciaro locality) and the eruptive apparatus of 1536. The surrounding area is characterized by an extensive shrub–herbaceous vegetation cover dominated by the *Genista* and *Ferula* genera, with the presence of isolated *Quercus ilex* trees and no arable land [35]. In the area, semi-wild cattle breeding is commonly practiced. The cave is inhabited by a large mixed colony of bats (approximately 600–700 Minioptera, along with several dozen large *Myotis* and Rhinolophids) [35].Grotta dei Pipistrelli (Cassano allo Jonio, CS, 39°47′11″ N, 16°18′28″ E, cadastral number: Cb CS 110). Large cavity consisting of a succession of caverns whose bottom is occupied by debris and guano. It hosts a breeding colony of bats with a prevalence of *Myotis myotis* and *Miniopterus schreibersii*. The territory in which the cavity opens is hilly in the western part and slopes down towards the Piana di Sibari in the East, characterized by the presence of agricultural activity.Grave Grubbo (Verzino, KR, 39°15′41.4″ N, 16°51′45.1″ E, cadastral number: Cb KR 258), opens at 285 m a.s.l. and extends for 1926 m; the cave belongs to the extensive Le Grave Complex, the second longest system developing in gypsum deposits in Italy. The cave is active and water flows into one of its branches from an inlet point during rainy periods. The cave hosts a large, mixed breeding colony of bats (Minioptera and Rhinolophids). The surrounding area is characterized by extensive crops and olive groves, as well as cattle and sheep breeding.

#### 2.1.2. Handling and Sampling

The capture is an unavoidably invasive technique and was carried out only by expert and authorized personnel, according to current legislation (Permit of Istituto Superiore per la Protezione e la Ricerca Ambientale (ISPRA) Prot. N. 14589/T-A31). When possible, the bats were caught by hand in order to minimize the risk of disturbance to the colony (Grotta Chiusazza, Grotta Burrò, and Grave Grubbo) [36]. In some cavities a hand-held net with a telescopic handle was used to capture static and non-flying bats (Grotta Chiusazza, Grotta Burrò and Grave Grubbo). Only in 2 cavities (Grotta dei Pipistrelli of Sortino and Grotta dei Pipistrelli of Cassano allo Ionio) was it possible to use the harp-trap to catch specimens that flew off in the exit from the cave. After capture, each bat was placed for a short time inside a numbered canvas bag whose mouth was closed by a cord. Each bag contained only one subject. All operations, including the safety devices used by the operators, complied with the Guidelines for the monitoring of Chiroptera drawn up by the Istituto Nazionale Fauna Selvatica [1].

Rectal, conjunctival, and oral swabs for bacteriological survey were obtained from each bat using individually packed sterile microbiological swabs moistened with sterile saline solution 0.9%, inserting the tip and gently rotating it against the mucosa. The swabs were subsequently inserted into tubes containing Amies transport medium (Copan Italia, Brescia, Italy) and kept in a cooler with frozen gel packs for purposes of transport for a maximum of 8 h before culture-plate inoculation, or further storage in a refrigerator at 4 °C for a maximum of another 24 h, if no earlier processing was possible due to logistical reasons. All bats were handled for the shortest time possible and were released immediately after sampling.

### 2.2. Bacterial Isolation and Identification

The samples were transported in conditions of refrigeration to the laboratory and examined to detect the presence of potential pathogens. All samples (n. 567) were examined for Gram-negative bacteria; conjunctival and oral swabs (n. 378) were also submitted to bacteriological examination for Gram-positive bacteria. Rectal swabs (n. 189), after an enrichment in buffered peptone water, were streaked into MacConkey agar plates (Biolife Italiana, Milano, Italy). Conjunctival and oral swabs were cultured in nutritive broth and, after, streaked into MacConkey agar plates and into Staphylococci 110 Medium plates (Biolife Italiana, Milano, Italy). Colonies demonstrating distinctive macroscopic appearance were considered separate organisms and isolated on new plates. Isolates were subcultured in blood agar plates for identification by mass spectrometry MALDI-TOF (matrix assisted laser desorption/ionisation–time of flight mass spectrometry). The isolated colonies were seeded in a 48-well metal plate with disposable loops, using as a reference strain Escherichia coli ATCC 8739. The spectra were analysed by VITEK MS system (bioMérieux SA, Marcy l’Etoile, France) using the software Axima (Shimadzu Kyoto, Japan)-SARAMIS database (Spectral Archive and Microbial Identification System) (AnagnosTec, Berlin, Germany). Eighty-eight strains, unidentified by MALDI-TOF mass spectrometry, after being grown on blood agar base (Biolife Italiana, Milano, Italy) and diluted in physiological solution, were typed at the Laboratory of Specialized Bacteriology of the Zooprophylactic Institute of Sicily, using the traditional macro test tube method [37,38,39]. The bacteria of the genus *Bacillus* spp. (POS BAT 19/Rev 0) have been characterized by carbohydrates oxidation and fermentation, motility, urease, gelatinase, nitrate reduction, and Voges Proskauer (VP) tests; *Staphylococcus* and *Streptococcus* spp. (POS BAT 05/Rev 0 and POS BAT 30/Rev 0) were characterized by catalase, hemolysis, coagulase, oxidase, VP tests, and carbohydrate fermentation. The enterobacteria and Gram-negative glucose nonfermenting bacteria (POS BAT 09/Rev 0) were identified by oxidation–fermentation, mobility, catalase, oxidase, urease and tryptophanase tests, and utilization/fermentation/oxidation of carbohydrates. The serological typing of *Salmonella* spp. strains (POS BAT 04/Rev.4) was performed following the Kauffmann–White–Le Minor method, in agreement with the National Salmonellosis Center of Padua, Italy [40].

### 2.3. Statistical Analysis

We evaluated the difference in the number of strains belonging to pathogenic species isolated from different sampling sites and different bat species using Fisher’s exact test, using a significance level of *α* = 0.05. The data were analysed using RStudio Version 1.0.153 for macOS (https://github.com/rstudio/rstudio, accessed on 21 September 2022). The built-in function fisher.test was used to calculate the *p*-values.

## 3. Results

In 64 samples out of 378 tested (16.9%), the coexistence of Gram-positive and-Gram negative bacteria was found (12/189 conjunctival swabs (6.3%); 52/189 oral swabs (27.5%)).

### 3.1. Gram-Negative Strains

Four hundred thirteen Gram-negative strains were isolated from 567 tested samples. Of these, 377 belonged to 17 different genera of the family Enterobacteriaceae and 30 to five other families (Figure 1).

The most isolated species were *Enterobacter cloacae* (71 strains, 17.2%), *Hafnia alvei* (47 strains, 11.4%), *Citrobacter* spp. (46 strains, 11.1%), and *Serratia marcescens* and *Citrobacter freundii* (21 strains, 5.1%). Potentially pathogenic species including *Salmonella enterica*, *Klebsiella pneumoniae*, and *Pseudomonas aeruginosa* have also been identified. Only seven genera out of 23 (*Citrobacter*, *Enterobacter*, *Escherichia*, *Hafnia*, *Klebsiella*, *Morganella*, and *Providencia*) were detected at all three sampling sites (rectus, eye, and mouth). In most samples (294; 83.5%) a single bacterial strain was isolated. Samples in which two or three strains were isolated were few, 55 (15.6%) and 3 (0.9%), respectively (see Data Availability Statement). Figure 1B and Figure 2 show the results of bacteriological tests for the species of bat and the sampling site, respectively. Six strains have not been identified.

Only three species (*Enterobacter cloacae*, *Morganella morganii*, and *Serratia marcescens*) were present in all six caves. No significant difference was found in the number of strains belonging to pathogenic species isolated from different sampling sites. Six bacterial species (*Citrobacter freundii*, *Hafnia alvei*, *Klebsiella oxytoca*, *Morganella morganii*, *Serratia liquefaciens*, and *Serratia marcescens*) were present in all the bat species. Among potentially pathogenic species, the most frequently isolated were *Escherichia coli*, *Serratia marcescens*, and *Pseudomonas aeruginosa*. Table 3 summarizes the comparison of the percentages of pathogenic strains found in the different species of bats and *p*-value of significant differences calculated using Fisher’s Exact Test.

*Escherichia coli* is more common among *R. hipposideros* than in *M. myotis*, *M. schreibersii*, and *M. cappaccinii*. Fisher’s exact test showed significant differences between *R. hipposideros* and *M. schreibersii*. *Serratia marcescencens* occurs more frequently in *M. cappaccinii* than in *M. myotis*, *M. schreibersii*, and *R. hipposideros*. The same test also showed significant differences between *M. cappaccinii* and *M. myotis* as well as *M. cappaccinii* and *M. schreibersii*. *M. myotis* is characterised by a higher occurrence of *Pseudomonas aeruginosa* compared to *M. schreibersii*, *R. hipposideros*, and *M. cappaccinii*. *P. aeruginosa* levels are also significantly higher in *M. myotis* than *M. schreibersii* and *R. hipposideros*.

### 3.2. Gram-Positive

One hundred eighty-three Gram-positive strains belonging to six different genera were isolated from a total of 378 samples (Figure 3). Of these, 73 belonged to the Staphylococcaceae family, 72 to the Bacillaceae family, and 36 to four other families. Two strains have not been identified. The most frequently isolated species were *Enterococcus faecalis* (28 strains, 15.3%), *Bacillus licheniformis* (15 strains, 8.2%), *Bacillus megaterium* (14 strains, 7.7%), and *Staphylococcus sciuri* (12 strains, 6.6%). Only one strain belonging to a potentially pathogenic species (*Staphylococcus aureus*) was found in a specimen of *Myotis myotis*.

Figure 3B and Figure 4 show the results of bacteriological tests for the species of bat and the sampling site, respectively. Bacteria belonging to the genera *Bacillus* and *Staphylococcus* were present in all the caves, but the only common species found was *Enterococcus faecalis*. The bacterial species found in all four species of bats were *Enterococcus faecalis*, *Bacillus cereus*, and *Staphylococcus epidermidis*. In 11 samples (6.4%), two different strains were isolated (see Data Availability Statement).

## 4. Discussion

In recent years, a new field of research has emerged that outlines the importance of the microbiota in health and disease. The microbiome plays a key role in host evolution and can significantly contribute to various functions of the organism, such as sugar metabolism [41], digestion and the absorption of nutrients [42,43], and the production of metabolic enzymes [44].

We investigated the composition of the cultivable microbiome of Southern Italian Troglophile bats, using a culture-based approach. The present study also investigates the presence of potentially pathogenic Gram-negative and Gram-positive bacteria. There is a growing awareness of how the spread of pathogens in wild animals can impact human and animal health. The emergence of new infectious diseases is not only a conservation issue, due to the dangers it poses to protected species, but also a potential threat to public health. The isolation of infectious agents in bats indicates the importance of monitoring the presence of potentially pathogenic bacteria in this order, isolated from different bat populations.

Our analysis led to the identification of approximately six hundred bacterial strains belonging to ninety different bacterial species (50 Gram-negative and 40 Gram-positive) and 29 genera (Figure 1 and Figure 3). Our results demonstrate the presence of a wider variety of bacterial species than indicated by previous research on European bats. Di Bella et al. (2003) [17] isolated 26 bacterial species belonging to 13 genera from faecal samples (*Acinetobacter*, *Alcaligenes*, *Citrobacter*, *Enterobacter*, *Escherichia*, *Hafnia*, *Klebsiella*, *Kluyvera*, *Morganella*, *Proteus*, *Pseudomonas*, *Streptococcus* (now *Enterococcus*), and *Yersinia*), all of which were also found in our survey, with the exception of *Yersinia* spp. Some of the species isolated in the present study can be considered commensals or environmental contaminants (e.g., *Bacillus* spp.), but others (e.g., *Salmonella enterica*, *Klebsiella pneumoniae*, *Pseudomonas aeruginosa*, and *Staphylococcus aureus*) are potentially pathogenic, both for bats and for other animals and also for humans [9,45]. *Enterococcus faecalis* was the most isolated species among Gram-positive bacteria, according to other authors [46]. It has been held responsible for several diseases in bats (septicemia; pneumonia; myocarditis; and wound infection) [46].

Such bacteria could probably be endemic to bats and play a mutually beneficial role, providing the host with stable growing conditions and additional nutrients [47]. For example, we have found the presence of *Serratia marcescens* in the oral cavity of all four species of bats examined. Galizia et al. (2014) [47] found *Serratia marcescens* in the oral cavity of hematophagous bats but not in frugivorous and nectivorous, hypothesizing a role of *Serratia* in the production of an extracellular protein that binds to the hemo-*has*A group, which allows the release of the heme group from hemoglobin and thus acquire the iron necessary for its metabolism. This bacterial species had been isolated by other authors in the intestine of frugivorous [48,49]. *Shigella* spp. are rarely detected in animals other than primates. The presence of these bacterial species may be due to water or food source contamination, or also through transmission from other bats within colonies [33]. In addition, the high number of species isolated from faecal samples is potentially related to the presence of endemic bacteria in ingested insects [49]. Habitat preference, geographic origin, and eating habits are factors that can influence the colonization of bats with different bacteria. Our results indicate a highly varied distribution of bacterial species in the different sampling sites (only three species in common for Gram-negative and one species for Gram-positive) but no significant difference was found in the distribution of pathogenic species.

The most commonly isolated bacterial genus in our study was Enterobacter, which has the ability to break down a large number of sugars [41]. Five bacterial genera (*Citrobacter*, *Hafnia*, *Klebsiella*, *Morganella*, and *Serratia*) were isolated from all bat species. Hafnia produce, among others, enzymes that allow the degradation of chitin, favouring digestion and the assimilation of nutrients in insectivorous bats [42]. The Serratia genus also appears to play an active role in food degradation in several bats [43].

It would be useful to compare our results with data on the microbiota of further species and populations living in other regions that have habitats and eating habits similar to the subjects of our study. Further studies are needed to identify any bacterial species that constitute a common core of troglophile bat microbiomes and to evaluate which species play an important physiological or nutritional role.

Several bacteria detected in our investigation have also been found by other authors in Europe, reinforcing the hypothesis of a common bacterial core [17,44]. Furthermore, Hughes et al. (2018) demonstrated that there are no distinct shifts in microbiome composition between adult and juvenile insectivorous bats; the authors hypothesize that, through the production of specific enzymes, this core of bacteria contributes to the fast metabolism necessary to rapidly provide energy for flight [44].

The variety of bacterial species that make up the intestinal microbiome of insectivorous bats depends on the environment, diet, and numerous other factors [21]. Experimental studies have shown that the microbiomes of insectivorous bats placed in captivity converge within a six-week period [50]. We believe it is important to continue studying the microbiome of insectivorous bats from other regions with similar habitats to verify the hypothesis that there is a symbiotic microbial core that has evolved hand in hand with the hosts.

## 5. Conclusions

This study provides novel data on the cultivable microbiome of Southern Italian bats, revealing a greater diversity of bacterial strains than measured by previous studies. Several physiological bacteria were found to be common to all the studied bat species. These strains may have evolved hand in hand with their hosts and thus constitute a common core of symbiotic bacteria, but further studies are needed to evaluate their biological and ecological functions. The results of the present study also indicate the presence of pathogenic and potentially pathogenic strains in wild bat populations in Italy, but not in alarming numbers. There is no evidence that the bats examined constitute a spread hazard of zoonotic bacterial agents we researched. Nevertheless, troglophile bats populations can be considered a good indicator of environmental contamination by potentially pathogenic bacteria and should be regularly monitored for conservation and public health purposes.

## Figures and Tables

**Figure 1 animals-12-02684-f001:**
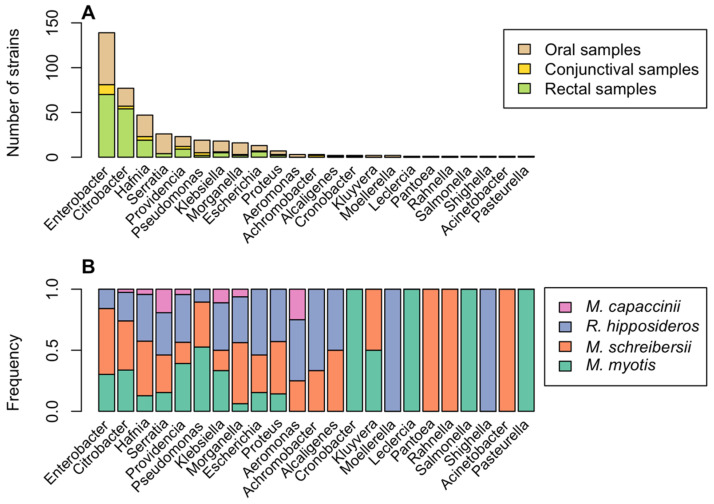
**Occurrence of Gram-negative bacteria.** (**A**) Number of Gram-negative bacterial isolates belonging to different 23 genera found in oral (light brown), conjunctival (yellow), and rectal (green) samples. (**B**) For each genus, the percentage of isolates found in different bat species is indicated: *Myotis myotis* (cyan), *Miniopterus schreibersii* (orange), *Rhinolophus hipposideros* (blue), and *Myotis capaccinii* (pink).

**Figure 2 animals-12-02684-f002:**
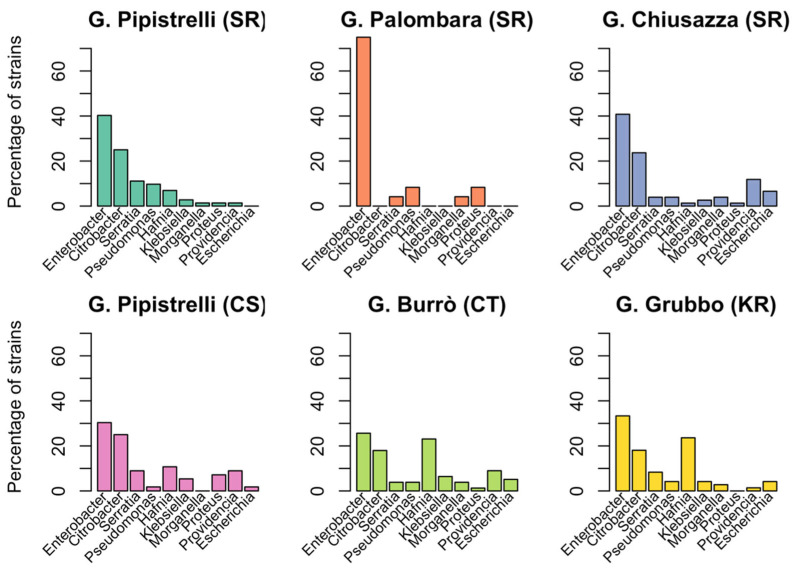
**Distribution of Gram-negative bacteria in different sampling locations.** The percentage of isolates belonging to the 10 genera of Gram-negative bacteria most commonly occurring in our dataset is shown for 6 different sampling sites: Grotta dei Pipistrelli, SR (cyan), Grotta Palombara, SR (orange), Grotta Chiusazza, SR (blue), Grotta dei Pipistrelli, CS (pink), Grotta Burrò, CT (green), and Grave Grubbo, KR (yellow).

**Figure 3 animals-12-02684-f003:**
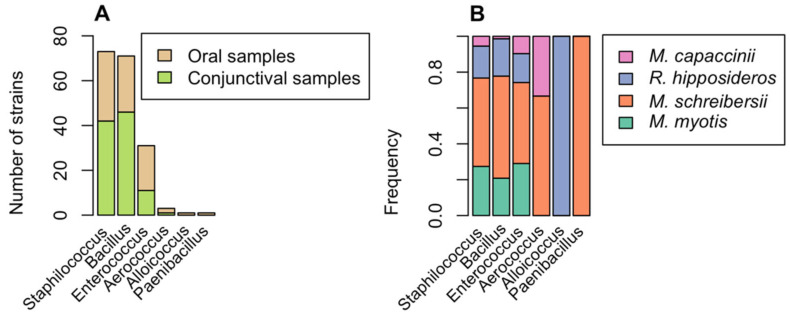
**Occurrence of Gram-positive bacteria.** (**A**) Number of Gram-positive bacterial isolates belonging to different six genera found in oral (light brown) and conjunctival (green) samples. (**B**) For each genus, the percentage of isolates found in different bat species is indicated: Myotis myotis (cyan), Miniopterus schreibersii (orange), Rhinolophus hipposideros (blue), and Myotis capaccinii (pink).

**Figure 4 animals-12-02684-f004:**
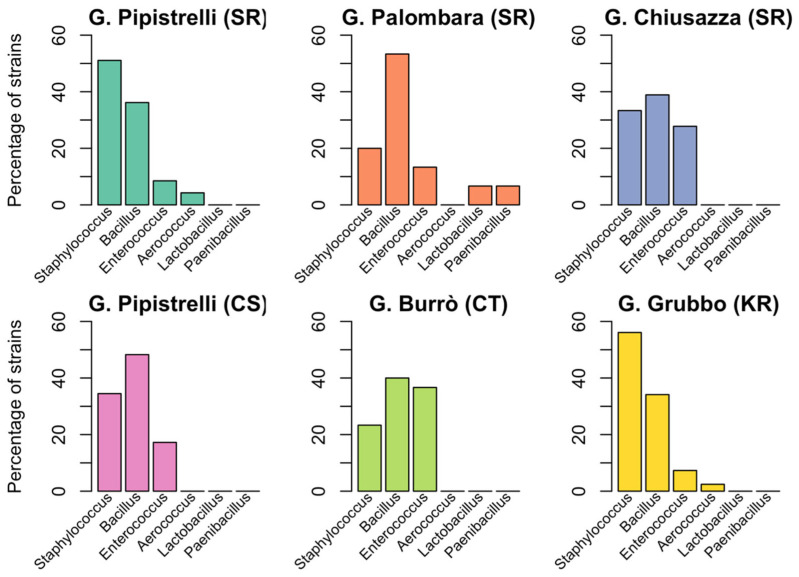
**Distribution of Gram-positive bacteria in different sampling locations.** The percentage of isolates belonging to six genera of Gram-negative bacteria is shown for six different sampling sites: Grotta dei Pipistrelli, SR (cyan), Grotta Palombara, SR (orange), Grotta Chiusazza, SR (blue), Grotta dei Pipistrelli, CS (pink), Grotta Burrò, CT (green), and Grave Grubbo, KR (yellow).

**Table 1 animals-12-02684-t001:** Individuals sampled.

Superfamily	Family	Subfamily	Species	n. Individuals
Vespertilionoidea	Vespertilionoidae	Myotinae	*Myotis myotis*	47
			*Myotis capaccinii*	8
	Miniopteridae		*Miniopterus schreibersii*	91
Rhinolophoidea	Rhinolophidae		*Rhinolophus hipposideros*	43
Total				189

**Table 2 animals-12-02684-t002:** The number of sampled individuals in the 6 study areas by species.

Site	Number of Sampled Individuals				
	*Myotis myotis*	*Miniopterus schreibersii*	*Rhinolophus hipposideros*	*Myotis capaccinii*	Total
Grotta dei Pipistrelli (SR)	13	12	17	8	50
Grotta Palombara (SR)	5	9			14
Grotta Chiusazza (SR)	12	13	7		32
Grotta dei Pipistrelli (CS)	16	15			31
Grotta del Burrò (CT)	1	15	17		33
Grave Grubbo (KR)		27	2		29
Total	47	91	43	8	189

**Table 3 animals-12-02684-t003:** Comparison between the percentages of pathogenic strains isolated in different bat species and *p*-value of significant differences, calculated with the Fisher’s exact test.

	MS/MC	MS/MM	MS/RH	MC/MM	MC/RH	MM/RH
*Escherichia coli*	4.4%/0%	4.4%/4.3%	4.4%/16% *p* = 0.0374	0%/4.3%	0%/16%	4.3%/16%
*Serratia marcescens*	7.7%/50% *p* = 0.0048	7.7%/6.4%	7.7%/16.3%	50%/6.4% *p* = 0.0059	50%/16.3%	6.4%/16.3%
*Pseudomonas aeruginosa*	3.3%/0%	3.3%/21.3% *p* = 0.0012	3.3%/0%	0%/21.3%	0%/0%	21.3%/0% *p* = 0.0012

Legend: MS = *Miniopterus schreibersii*; MC = *Myotis cappaccinii*; MM = *Myotis myotis*; and RH = *Rhinolophus hipposideros*.

## Data Availability

The data that support this study are available in Mendeley Data repository at doi: 10.17632/tct4zh9fy6.1].

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
