# Peer review of "Cultivable Bacteria Associated with the Microbiota of Troglophile Bats"

_animals, 2022, doi:10.3390/ani12192684_

Round 1

Reviewer 1 Report

It is a great sampling effort and also gives knowledge about the bacteria found in the different regions.

I would change the title so that it is not biased and against Bat Conservation, something like:

Cultivable bacteria associated with the microbial flora in troglophile bats

The authors indicate that the issue of the relationship between viruses and bats has been little explored. A simple search on google scholar turns up at least a hundred related articles on the matter.

The conclusion in the abstract is dangerous and not very robust. Scientists dedicated to bats know that this exists, it is a conclusion that bats can be attacked.

Perhaps the findings can tell us about the symbiotic relationship with bats that these GRAM bacteria make there.

I understand that the researchers are only talking about cultivable bacteria. Today there are new generation techniques that give us a little more knowledge. Using a simple extraction kit they can analyze the Ribosomal V4 Region and obtain a wide range of host-associated microbiota if they are interested in knowing about bacteria. Perhaps this method can be complementary to this cultureable bacteria method.

Do not forget the bibliographic citations an ex. On lines 52-53

All Italian Chiroptera feed on Arthropods and are nocturnal even if several species come 52 out of shelters in the presence of light. The diet of Myotis capaccinii also includes aquatic 53 larvae of Diptera and fish fry.  CITA

The sites of the caves, it is necessary to put geographical coordinates. Relevant biological and meteorological data.

The authors do not indicate how many sampling nights and the hours for obtaining samples

They have a collection permit; they respected the ethical treatment of the animals. They used face masks for the captures, etc.

The data is very good and the authors could do something very nice if they focus and revise the theory of the holobiome, microbiota, etc. Bacteria are not only bad, in the last paragraph they say they can be of help to bats because they are ancient associations.

Reviewer 2 Report

The study of the microbiome of bats is of great interest in the One Health concept since most of these species are considered sentinels of the health status of ecosystems. It includes the analysis of a large number of samples from different locations in Italy, which allows the comparison of some populations with others, as well as between species.

However, the manuscript is poor. It is essential to review it thoroughly to correct the English, as well as editing issues that appear throughout the text (size and type of font, bold, double spaces, merged words, etc).

The introduction is not bad, except for the flaws in the English translation. The purpose of the study is constantly changing. In the simply summary, the objective is to evaluate epidemiological factors of the flora (please, do not use the term flora, use microbiota) and to show the presence of pathogens (you mean to investigate... at this point in the study we still do not know if they are carriers of pathogens, although they are expected to be). In the abstract and in the manuscript, the objective improves, adapting better to the content of the manuscript.

Important tips:

- When we use acronyms we must always indicate what they mean the first time they appear in the text, as you have done with SARS, MERS... However, the meaning of NIH (L.76) does not appear, nor does that of OF ( L.171).

- Gram-positive and Gram-negative are always linked with a hyphen.

- The cardinal points in English are written with the first letter capitalized (Southern Italy).

- In statistics, the p-value is represented with lower case (p = 0.05).

The material and methods need to be improved. The sampling section should be divided into two: "Study population" and "Handling and sampling". Thus, all the information about the species included in the study can be written in the first one (in the manuscript the habitat of only one of them is mentioned, the rest is only mentioned about feeding). Perhaps it would be interesting to analyze if the four species share feeding/flight areas, if they coincide during activity hours or if they interact with each other in general. In this section, you can also add information about the locations where the samples have been collected. The "Handling and sampling" section would be used to describe the sample collection step by step. In this way, everything is not mixed up, since now it is written in a chaotic style.

L. 139. The samples are preserved in saline 09%, could it be that a point is missing (0.9%)?

The description of the statistical analysis is very poor, it is necessary to develop a little more. It remains to add the software used, commercial house, etc. In my opinion, I believe that the Fisher test should have been complemented with a Generalized Linear Model (GLM) in order to better assess the differences between species.

The description of the results is very dense and difficult to read. I think there is too much information in tables that is likely to tire the reader out and get lost. It would be interesting to change some tables for figures.

The discussion is very very poor... in a scientific article, the discussion is one of the most important sections where the results obtained are compared with those of other colleagues. It would also be interesting to discuss the statistical differences found between the different species: could it be due to their diet? for his behavior? Can one species be more urban than another? Half a page of discussion is not enough for the results you have obtained.

Finally, the conclusions only focus on pathogens, giving the reader the feeling that after all this analysis the only important thing is that bats are carriers of pathogens. In my opinion, the conclusions must be completely reformulated and the microbiological diversity obtained should be highlighted.

Round 2

Reviewer 1 Report

The manuscript has improved a lot and is more logical, the title would only remove the word pysiological (it doesn't make much sense)

The authors followed all recommendations,

What are the authors referring to with physiological microbiota?

I would remove physiological

I think that the title as simple as that is better, in the discussion they will be able to discuss if the results obtained have to do with the physiology or not of the individual

Cultivable bacteria associated with the microbiota of troglophile bats

Check Table 3. Upper case comparison

Reviewer 2 Report

The quality and writing of the article has greatly improved. In addition to the suggested changes, the authors have reformulated the content, giving a better view of the results presented.

The methodology is more orderly, clear and detailed, the new figures that replace the tables of the first draft are very visual and informative, which is appreciated, and the discussion has been completed.

There are only a few small details that need to be modified:

When you mention Enterobacteriaceae Group, group would go without a capital letter, although my advice is to change group to family, since on other occasions you talk about families. Check lines 34 and 285.

When writing the text, numbers less than 10 are written (one, two, three...) while the others are kept in numbers (74, 87, 134...). Check lines 34, 36, 285, 299, 300, 302, 333, 335, 342, 350, 355, 391 and 404.

There are still some punctuation errors, double spaces, italics that shouldn't be, wrong upper or lower case, etc. Please check lines 87, 136, 254, 297, 320, 338, 372 and 387.

There are also some cardinal points in lowercase. Please check lines 91, 92, and 103.

In English, the common names of animal species are written in lower case, unless they derive from a proper name or geographic region. Please correct lines 102, 116, 129, and 141.

In lines 150 to 218, please justify the text.

Line 222, please remove the name of the authors of the text (RG and MTS), it is not scientifically correct.

Line 295: "Six strains have not been identified." It is usually better to start with what has been identified and at the end indicate what has not been achieved. I suggest moving this sentence to the end of the paragraph.

Line 368: "gaining increasing interest" sounds redundant, please rephrase.
